# Immunotoxicity In Vitro Assays for Environmental Pollutants under Paradigm Shift in Toxicity Tests

**DOI:** 10.3390/ijerph20010273

**Published:** 2022-12-24

**Authors:** Xinge Wang, Na Li, Mei Ma, Yingnan Han, Kaifeng Rao

**Affiliations:** 1Key Laboratory of Drinking Water Science and Technology, Research Center for Eco-Environmental Sciences, Chinese Academy of Sciences, Beijing 100085, China; 2State Key Laboratory of Environmental Aquatic Chemistry, Research Center for Eco-Environmental Sciences, Chinese Academy of Sciences, Beijing 100085, China; 3National Engineering Research Center of Industrial Wastewater Detoxication and Resource Recovery, Beijing 100085, China; 4University of Chinese Academy of Sciences, Beijing 100049, China

**Keywords:** adverse outcome pathway, high-throughput, immunosuppression, skin sensitization

## Abstract

With the outbreak of COVID-19, increasingly more attention has been paid to the effects of environmental factors on the immune system of organisms, because environmental pollutants may act in synergy with viruses by affecting the immunity of organisms. The immune system is a developing defense system formed by all metazoans in the course of struggling with various internal and external factors, whose damage may lead to increased susceptibility to pathogens and diseases. Due to a greater vulnerability of the immune system, immunotoxicity has the potential to be the early event of other toxic effects, and should be incorporated into environmental risk assessment. However, compared with other toxicity endpoints, e.g., genotoxicity, endocrine toxicity, or developmental toxicity, there are many challenges for the immunotoxicity test of environmental pollutants; this is due to the lack of detailed mechanisms of action and reliable assay methods. In addition, with the strong appeal for animal-free experiments, there has been a significant shift in the toxicity test paradigm, from traditional animal experiments to high-throughput in vitro assays that rely on cell lines. Therefore, there is an urgent need to build high-though put immunotoxicity test methods to screen massive environmental pollutants. This paper reviews the common methods of immunotoxicity assays, including assays for direct immunotoxicity and skin sensitization. Direct immunotoxicity mainly refers to immunosuppression, for which the assays mostly use mixed immune cells or isolated single cells from animals with obvious problems, such as high cost, complex experimental operation, strong variability and so on. Meanwhile, there have been no stable and standard cell lines targeting immune functions developed for high-throughput tests. Compared with direct immunotoxicity, skin sensitizer screening has developed relatively mature in vitro assay methods based on an adverse outcome pathway (AOP), which points out the way forward for the paradigm shift in toxicity tests. According to the experience of skin sensitizer screening, this paper proposes that we also should seek appropriate nodes and establish more complete AOPs for immunosuppression and other immune-mediated diseases. Then, effective in vitro immunotoxicity assay methods can be developed targeting key events, simultaneously coordinating the studies of the chemical immunotoxicity mechanism, and further promoting the paradigm shift in the immunotoxicity test.

## 1. Introduction

For all living organisms, various physiological functions require constant surveillance and regulation of the immune system. Different from other systems, the immune system distributes throughout the body, and establishes close connections with other systems, including the respiratory system, the digestive system, the reproductive system and the skin. Therefore, the immune system may be a highly vulnerable toxic target, regardless of the route of exposure to environmental pollutants. In the *Guidance for immunotoxicity risk assessment for chemicals*, the world health organization (WHO) pointed out that the incidence of allergic and autoimmune diseases has increased in the past few decades, and that infectious diseases and tumors are still a major public health burden. Studies have shown that the influence of environmental pollutants on the immune system may be one of the most important reasons for this increase [1,2,3]. Besides immunity-related diseases, other toxic endpoints may also be associated with immune system damage. For example, liver toxicity caused by the persistent organic pollutant perfluorooctanesulfonic acid (PFOS), may be associated with inflammatory cell infiltration [4]. Some pollutants can also affect the immune system of aquatic organisms, resulting in the large-scale spread of infectious diseases [5] and serious damage to ecological health. However, the immunotoxicity of environmental pollutants has not received enough attention, and there is a lack of relevant research and data.

With the continuous increase in chemical types and quantities, a large number of harmful substances are produced and released into the environment, posing a threat to human beings and other organisms. Traditional in vivo experiments can no longer meet the needs of chemical toxicity assessments. Therefore, Tox21 was proposed in 2007; it appealed to scientists to develop an in vitro assay using the primary cell or cell line, and this indicated that the toxicity test would experience a significant paradigm shift so that high-throughput non-animal experiments, with short cycle, low cost and convenient operation, would gradually replace traditional animal experiments (Tox21). In addition, the proposal of an adverse outcome pathway (AOP) further points to a feasible direction for a paradigm shift in the toxicity test. Many international agencies, including the Organization for Economic Co-operation and Development (OECD) and U.S. Environmental Protection Agency (EPA), have made great efforts to promote the development of in vitro assay methods, especially methods for genotoxicity and estrogenic effects, which are relatively mature and widely used. For example, the Ames test and MCF-7 cell proliferation assay provide excellent conditions for a paradigm shift in genotoxicity and endocrine disruption, respectively. In recent years, our team has also been committed to improving current in vitro assay methods, establishing new methods and applying them to the detection of environmental samples, and has achieved remarkable results [6,7,8,9]. In fact, different toxicity targets have different sensitivities, which profoundly affect the hazard threshold of pollutants. In addition, studies have shown that the immune system is more sensitive than other toxic endpoints, thus it is necessary to evaluate the immunotoxicity of environmental pollutants.

Up to now, immunotoxicity studies are still based on conventional animal experiments, lagging far behind the genotoxicity and endocrine disrupting effect. However, considering the trend in the paradigm shift regarding the toxicity test, many studies have made attempts to search for more convenient and less time-consuming ways to utilize primary immune cells or cell lines, such as the immunosuppression detection method using Mishell–Dutton culture [10], and functional assays of various immune cells [11]. However, due to the complexity of the immune system, there are many problems and challenges we are faced with, which mainly include complex and diverse detection indexes, unclear mechanisms of action, incomplete AOPs, and even ambiguous judgments of immunotoxicity. 

In fact, due to the particularity of the immune system, medical scientists have made far more contributions than environmental scientists to the study of immunotoxicity. In this study, we try to learn some methods or ideas from medicine or immunology and apply them to the environmental field. With the ultimate aim of developing high-throughput methods, this paper summarizes in vitro methods derived from guidance documents developed by international organizations (e.g., OECD), methods used in the scientific literature, and methods described in chemical databases (e.g., Pubchem), providing reference for the future application of immunotoxicity detection methods in the environmental field.

## 2. Immunotoxicity

As defined by the U.S. EPA, immunotoxicity is the ability of a test substance to suppress an immune response that can increase the risk of infection or tumor disease, or cause an inappropriate stimulation of the immune system that can lead to allergies or autoimmune diseases (OPPTS 870.7800, 1998). Exogenous substances that interfere with the immune system can be natural toxic chemicals or drugs, but in the field of the environment, we are often concerned with artificial chemicals that are intentionally or unintentionally released into the environment during industrial production and life. According to human clinical adverse reactions, immunotoxicity can be divided into four categories: immunosuppression, immunostimulation, hypersensitivity and autoimmunity [12]. Immunosuppression and immunostimulation are collectively referred to as direct immunotoxicity [13], which are manifested as the inhibition or stimulation of immune cells or their functions in immune tissues, including phagocytic changes in cytokines produced by lymphocyte proliferation. A case in point is HIV, an infectious disease virus that specifically attacks cluster differentiation (CD)4+T cells, resulting in the collapse of the human immune system and eventually death from other complications. Immunostimulation may be beneficial to organisms to some extent, but here it refers to excessive or inappropriate stimulation that can interfere with normal defense mechanisms, making them respond in ways harmful to the host. Hypersensitivity reaction and autoimmune reaction belong to indirect immunotoxicity, which are the result of immunostimulation, to a certain extent. In essence, they are specific immune reactions that damage the body, caused by exogenous chemicals or macromolecules modified by these substances. More specifically, a hypersensitivity reaction is caused by tissue damage or physiological dysfunction after the body is stimulated by the same allergen again, in which the antigen–antibody reaction is often the decisive process; however, that delayed hypersensitivity (DTH) is associated with T cell sensitization. Autoimmunity is a process in which the immune system attacks its own components due to dysfunction and regulatory disorder. Some exogenous substances can trigger autoimmunity by changing their own antigens, preventing the central tolerance of reactive T and B lymphocytes and changing gene expression [14]. In the process of inducing indirect immunotoxicity, substances that directly induce the immune response are called antigens, which can activate the proliferation and differentiation of immune cells, and produce specific antibodies or sensitized lymphocytes; these substances can also bind to these antibodies or lymphocytes. These two properties are called immunogenicity and immunoreactivity, respectively. However, for environmental chemicals, due to their small molecular weight and lack of immunogenicity, they generally cannot act as natural antigens and strongly stimulate immune responses [15]; however, they may harm the body by inducing non-immune responses that impair the immune system (direct immunotoxicity) or by binding to carrier proteins to trigger an immune response (indirect immunotoxicity) [16]. More complicated than direct immunotoxicity, indirect immune toxicity involves other organizations and systems of the body, mainly related to genetic background, and it is mainly concerned with food safety, cosmetic sensitization, disease treatment and so on. In fact, either type of immunotoxicity is the result of a disorder in the check and balance system, and is likely to cause further severe disease. 

## 3. Paradigm Shifts in Toxicity Tests

With the development of the economy, a large number of environmental chemicals have been produced and released into the environment, and traditional toxicity assay methods that rely on animals can no longer meet the demands of chemical management. In order to solve this problem, in 2007, the U.S. National Research Council issued a report, Toxicity Testing in the 21st Century: A Vision and a Strategy, and proposed new ideas for toxicological research and toxicity testing strategies. They pointed out that toxicity tests should be transferred from expensive and time-consuming animal experiments to the combination of in vitro assay and computational toxicology, simultaneously exploring the interaction between pollutants and biomolecules, and the disturbance mechanisms of key toxicity pathways. This marks the beginning of a paradigm shift in toxicity tests. In 2008, the Tox21 project was officially launched in the United States, based on high-throughput screening (HTS) technology, aiming to more quickly and efficiently determine the potential of certain chemicals to lead to negative health effects [17]; however, it greatly relied on mature and stable in vitro assay methods. In the early stage, HTS was only used for drug screening, which can be divided into several categories according to biological characteristics: receptor-binding assay, enzyme assay, cell viability assay, metabolite assay, gene expression assay, etc. These assays actually provide rich experiences and clear development ideas for HTS of environmental toxic chemicals, and the first step is to establish an effective and stable in vitro assay. Currently, Tox21 has adopted 232 different bioassays [18]. According to the experience of HTS for drugs, these assays can be divided into two categories: target based or cell phenotype-based methods [19]. Target-based methods target specific moleculars, such as receptors (e.g., estrogen receptors (ERs), androgen receptors (ARs) and aryl hydrocarbon receptors (AhRs)), enzymes (e.g., acetylcholinesterase (AChE), aromatase and cytochrome P450) and DNA, showing clear detection indicators, but greatly depend on a detailed relationship between the target and toxic effects. For pollutants with unclear toxicity mechanisms, cell phenotype-based methods provide a better choice, using different cells as screening tools, and mainly detecting cytotoxicity, function damage, certain functional proteins, secretions and changes in signaling pathways. In the new era of a paradigm shift in toxicity tests, these screening models at the cellular and molecular level lay a necessary foundation for automatic operation, making it possible to replace manual screening with computer-controlled HTS. 

However, for such in vitro assays, it is difficult to predict the toxic effects at an individual level when only considering cellular damage or functional change. To solve this problem, the U.S. EPA formally proposed an AOP, a conceptual framework which describes the interconnectedness between a molecular initiating event (MIE) and an adverse outcome (AO) via a series of key events (KEs) in 2010. Here, MIE refers to the initial interaction between a molecule and a biomolecule or biological system, such as protein binding, receptor binding, enzyme inactivation, DNA damage and so on. Additionally, AOs tend to have harmful effects associated with risk assessment at different levels of biological tissue structure. Previously, in vivo and in vitro experiments were mutually separate, and it is AOP that organically unites them together. For that test, the endpoints of in vivo and in vitro experiments often correspond to AOs and MIE or MIE related microscale KEs, respectively. In general, AOP connects toxicity events at different levels through causality and greatly simplifies the toxicity assessment process of chemicals, thus providing powerful theoretical support for the development of in vitro assay methods and the shift in the toxicity testing paradigm; this makes it possible to develop chemical risk management strategies based on the mechanism of toxicity. In this framework, some test endpoints can be measured and used to qualitatively predict their downstream events, which greatly promotes the development and application of chemical toxicity assessment and management.

For environmental pollutants, current HTS methods focus on genotoxicity and endocrine disrupting effects, and most of them are analysis methods with clear targets based on the explicit mechanism of exogenous substances. However, due to the complexity of the immune system, immunotoxicity detection of pollutants mainly relies on in vivo experiments and animal models, and methods for immunosuppression and hypersensitivity have been developed and validated [20]. As for immunotoxicity in vitro assay methods, scientists have also made some efforts using mixed or purified primary cells from animals, have focused on a phenotypic analysis of immune cells, but have ignored the molecular mechanism of toxicity. Up to now, no methods have been considered as effective and stable immunosuppression assay methods, and only skin sensitization has developed in vitro models that are completely independent of animals.

The current paper introduces attempts to create immunotoxicity assays from two aspects, skin sensitization and direct immunotoxicity, to further illustrate the necessity and challenges of a paradigm shift in the immunotoxicity test.

## 4. AOP-Based Sensitizer Screening

In indirect immunotoxicity, allergies account for about one-third of adverse drug reactions [21]. Allergic contact dermatitis (ACD) and respiratory allergies, induced by low-molecular-weight allergens termed haptens, are the most common form of human immunotoxicity, but far more chemicals have been identified as skin sensitizers than respiratory sensitizers [22]. In fact, there has been a series of relatively mature and validated methods for ACD, primarily depending on an explicit AOP (shown in Figure 1). This AOP begins with the binding of small electrophilic molecules to proteins, and then the molecules with a protein carrier can be identified and presented to T cells, which will then proliferate and differentiate rapidly into hapten-specific effector and memory T cells [23]. When the same sensitizers enter the body again, they will be quickly recognized by circulating T cells, and cause serious symptoms such as erythema, edema, vesicles and intense itching [24]. In recent years, with the rapid development of cosmetics industry, a large number of chemicals have been required to pass the sensitization test. Driven by the REACH, animal-based chemical tests will be reduced to a minimum or even banned [25], putting great pressure on this industry. Therefore, based on an AOP of ACD, many in vitro methods for Kes have been developed and established to predict skin sensitization of chemicals [16]. For respiratory allergic reaction, it shares overall similar processes with ACD: combination of electrophilic molecules with proteins, dendritic cell (DC) activation and T cell proliferation. The difference is that skin sensitizers preferentially bind to cellular proteins, while respiratory sensitizers preferentially react with serum proteins [26]. Two different AOs may be associated with different effector T cells (Th1 and Th2). In the meantime, similar screening methods for respiratory sensitizers are also gradually carried out.

### 4.1. Protein Binding Assay

Electrophilic skin sensitizers can covalently bind to the skin nucleophilic center and form a stable complex to activate the immune reaction after penetrating the skin. This binding is considered as the MIE and, therefore, protein-binding ability is a necessary indicator of the sensitizing potential of chemicals. The Direct Peptide Reactivity assay (DPRA), based on this MIE, has been designed for sensitizer screening. DPRA emphasizes the combination with cysteine and lysine polypeptides, although other amino acids containing nucleophilic heteroatoms (e.g., histidine, methionine, and tyrosine) may also react with electrophiles [27]. That is because their consumption is not correlated with the titer of the allergen [28]. DPRA has been recommended as an alternative method for skin the sensitization test by ECVAM in 2013 and accepted by OECD [29]. Lalko et al. found that most respiratory allergens were also reactive in DPRA, but they preferentially reacted with lysine polypeptides rather than cysteine, which provided thought on screening method for respiratory allergens [30].

Due to the lack of a metabolic system, DPRA cannot be used to screen for pre/pro-hapten, substances that can activate an allergic reaction only when metabolically activated. Gerberick et al. improved DPRA by adding horseradish peroxidase and hydrogen peroxide (HRP/P) oxidation systems to the co-incubation system, and providing biologically relevant activation [31]. This method was called a peroxidase peptide reactivity assay (PPRA). DPRA also has deficiency in false-negative problems when predicting some hydrophobic substances. In order to solve this problem, an amino acid derivative reaction assay (ADRA) was developed. ADRA was based on the same experimental principle as DPRA, but replaced polypeptides with simulated skin proteins of N-(2-(1-naphthyl)acetyl)-L-cysteine (NAC) and α-N-(2-(1-naphthyl)acetyl)-L-lysine (NAL). Such an ADRA reaction solution greatly decreases the precipitation rate of tested chemicals [32], and can be used to screen highly hydrophobicity sensitizers. Yamaga et al. later changed the composition of the reaction solution to increase the solubility of the hydrophobic substance and improve the reactivity, and finally developed an ADRA-organic solvent (ADRA-OS) reaction system, which could evaluate highly hydrophobic substances, even the octanol water distribution coefficient (logKow) > 6 [33]. In addition, the ADRA method solves well the problem of the co-elution of chemicals and nucleophiles, which often affects HPLC analysis of DPRA [34]. 

In order to apply the above methods to multi-constituent substances, ADRA fluorescence detection (ADRAFL) methods were developed. Avonto et al. identified sensitizers based on their ability to react with DCYA, dansyl cysteamine (a substance with fluorescent) [35]. Fujita et al. developed an ADRAFL method using the natural fluorescence of nucleophiles, which is based on the nucleophile structure, containing a highly planar and rigid structure [36]. The fluorescence-based method can directly analyze the adducts quantitatively, enabling the application of the HTS method.

### 4.2. Keratinocyte Assay

As the first target of sensitizers, keratinocytes (KC) convert non-specific exogenous stimuli into the production of cytokines (such as TNF-α, IL-18 and IFN-γ) [37]. The KE1 and KC activation, includes the release of pro-inflammatory cytokines (e.g., IL-18) and the induction of cyto-protective cellular pathways. Based on these effects, there are two corresponding assay methods, the ARE-Nrf2 luciferase assay and the inflammatory cytokine assay. 

The ARE-Nrf2 luciferase assay was performed based on the signaling pathway Keap1-Nrf2-ARE, which is the cell defense against oxidative stress injury. Nrf2, as a transcription factor, normally binds to the intracellular and oxidative stress sensor protein, Kelch-like ECH-associated protein 1 (Keap1). Skin sensitizers can react directly or indirectly with the cysteine residues of Keap1 protein, thereby promoting the release of Nrf2. The free Nrf2 enters the nucleus and binds with other small molecules to form a complex, which binds to the antioxidant response element (ARE), initiating transcription of downstream genes [38]. The KeratinoSens™ test method and LuSens Test method, using ARE-Nrf2 luciferase, have been adopted by OECD. Both of them use immortalized human KC lines containing luciferin genes, and the luciferin signals can reflect the activity of electrophilic substance to activate Nrf2 [39]. The main difference between these two methods is the source of ARE elements.

In the process of ACD, a specific antigen signal (i.e., hapten–protein complex) is not the only signal for the activation of langerhan cells (LC), but DCs of the epidermis and cytokines produced by KCs during initial exposure are also an important factor, even determining the strength of the sensitization effect. Thus, the irritant capacity of allergens may present an additional risk factor for the development of contact dermatitis, suggesting that irritant allergens may be more potent than non-irritant allergens [40]. Based on this, the effectiveness of sensitizers can be assessed by detecting cytokine expression of KCs. Sensitizers (not irritants) can increase expression of IL-1α in the mouse KC cell line HEL30 and IL-12 in human KCs. In addition, IL-18 plays a proximal role in skin sensitization induction by promoting the secretion of pro-inflammatory mediators and promoting the Th1 immune response, but not in irritant contact dermatitis or asthma [41,42]. Therefore, increased IL-18 is also used to distinguish contact allergens from respiratory allergens or irritants [43]. Vandebriel et al. also evaluated the possibility that the gene expression profile of the KC cell line HaCaT [44], after in vitro exposure, could be used to predict the sensitization potential of chemicals; the classification model achieved a 70% accuracy in screening sensitizers. They also point out significant differences in the oxidative stress pathway gene Hmox1 between sensitizers and irritants. Therefore, the analysis of the gene expression profile may be a choice in the screening of sensitizers.

### 4.3. DC Activation Assay

KE2 in this AOP is the activation of DCs, which present antigens to T cells, and modulate T and B cell mediated immune responses. The activation markers can be specific cell surface molecules (e.g., CD86, CD80, CD54 and MHC II), chemokines (e.g., CXCL8), cytokines (e.g., IL-1β, TGF and TNF-α), E-cadherin and Langerin expression [22]. Sources of DCs have been introduced in a DC maturation assay for immunosuppression assessment. The difference is that the DC activation assay for sensitizers has been standardized and adopted by OECD [45], including h-CLAT, U-SENS™, or interleukin-8 reporter gene analysis (IL-8 Luc test). 

The h-CLAT and U-SENS™ use cell lines THP-1 and U937, respectively, to detect changes in the expression of cell surface markers to reflect sensitization ability. CD86 is one of the typical markers of THP-1 and U937 activation, and CD45 also reflects THP-1 activation. Cytotoxicity assays are required to determine whether noncytotoxic concentrations can cause marker changes. The increase in IL-8 mRNA or IL-8 protein is considered another biomarker to distinguish sensitizers from non-sensitizers in MoDC, U937 or THP-1 [46]. An IL-8 Luc test was performed using a THP-1 derived IL-8 reporter cell line THP-G8, which is regulated by IL-8 and glyceraldehyde 3-phosphate dehydrogenase (GAPDH) promoters, carrying stable luciferase orange (SLO) and stable luciferase red (SLR) luciferase genes, respectively [46]. In this method, the fluorescence of luciferase substrates is used as an indicator of IL-8 and GAPDH activity after cell exposure to sensitizers, to quantitatively measure luciferase gene induction. In addition, Kim et al. used RAW264.7 mouse Mφs as a model and suggested that IL-1α and IL-1β may serve as potential biomarkers of skin sensitization [47]. Genomic analysis can also be applied to the DC activation assay. One example is the genomic allergen rapid detection (GARD), which is based on the MUTZ-3 cell line and uses genomic biomarkers to screen for skin sensitizers, respiratory sensitizers or irritants [37]. 

### 4.4. T Cell Activation Assay

KE3 in this AOP is the activation and proliferation of T cells, a process that requires two synergistic signals from APCs, namely the peptide-MHC complex and co-stimulatory molecules (e.g., CD80 and CD86) to bind to T cell surface T cell receptor TCR and costimulatory receptor (e.g., CD28), respectively. 

Cytokines derived from APCs (e.g., IL-1α, IL-1β, IL-18, and TNF-α) are also essential for T cell activation and Th2 polarization. Due to the complexity of this process, it is difficult to simulate the process completely in vitro. Equivalent alternatives for KE1-KE3 have been developed, but there is no effective alternative for KE4, so T cell activation is generally assessed indirectly by a local lymph node assay (LLNA) [48], gold standard in vivo method for skin sensitization. However, research on in vitro methods for detecting T cell activation or polarization is still ongoing, usually based on co-culture models of DC and T cells. Nagahata et al. used Jurkat T cell lines (IL-2P::Jurkat cells) that stably express luciferin downstream of the pro-inflammatory cytokine IL-2 promoter, and assess skin sensitivity by luciferin activity [48]. This method can simultaneously monitor skin sensitization and the immune suppression of chemicals. Richter et al. established an in vitro human T cell initiation assay (HTCPA), in which naive T cells were co-cultured with stimulated mo-DCs, CD28 antibodies and cytokines [49]. After specific antigen restimulation, specific T cells were identified by detecting the production of cytokines, such as IFN-γ and TNF-α. However, these methods cannot distinguish between respiratory sensitizers and contact sensitizers, whereas Chary et al. pointed out that the typical cytokine fingerprint of respiratory sensitizers is the increased release of IL-4, IL-5, IL-10 and IL-13, while contact sensitizers can increase the production of TNF-α and IFN-γ [50]. Therefore, in vitro experiments targeting T cell activation can be refined by cytokine profiling to distinguish skin sensitizers from respiratory sensitizers.

Chary et al. described that the difference between skin sensitizers and respiratory sensitizers lies in the T cell polarization type they induce (i.e., Th1 or Th2 responses) [50]. Th2 promotes the proliferation of B cells, eosinophils and mast cells, and enhances airway hyperresponsiveness by releasing cytokines such as IL-4, IL-5, IL-9 and IL-13. B cells can differentiate into plasma cells to produce and release allergen-specific IgE, which can bind to high-affinity IgE receptors (FcεR1) on the surface of mast cells and basophil, triggering their degranulation and releasing histamine and pro-inflammatory factors. However, the role of IgE in respiratory sensitization is still unclear. Therefore, making an accurate distinction between skin sensitizers and respiratory sensitizers needs more understanding and research on respiratory sensitization, and this will build a solid foundation for the screening of respiratory sensitizers.

### 4.5. Co-Culture and 3D Model

The skin consists of an outer epidermal layer of KCs and a lower dermal layer of fibroblasts, interspersed with DCs, constituting an important defense barrier of the human body [51]. 

In the process of ACD occurring, each step involves complex interactions of different cells. Although the above in vitro skin sensitizers screening methods had a good statistical relationship with LLNA data, they could not effectively represent in vivo conditions in most cases [52]. Therefore, some scientists are devoted to establishing co-culture/3D models. Hennen et al. co-cultured HaCaT with THP-1 to allow interaction occurring, and detected the sensitization ability of HaCaT by analyzing the differences in expression of CD86, CD40 and CD54 on THP-1 cells [53]. Cao et al. found that, compared with THP-1 cultured alone, THP-1 co-cultured with KCs expressed more CD86 and CD54 on their surface after being exposed to skin sensitizers [54]. Karri et al. also compared the effects of KCs (HaCaT) and monocyte (THP-1), cultured alone with their co-culture, on cytokine responses induced by important skin sensitizers [55]. In the co-culture model, the secretion of IL-1β and IL-8 was significantly triggered by all tested haptens, while non-sensitizers and stimulants were not. The result indicated that, in the co-culture model, cytokine induction was more reliable and had higher levels. Overall, the co-culture model can not only distinguish sensitizers from stimulants, but can also improve the sensitivity of the test and reduce the concentration of chemicals required to achieve positivity. Lee et al. proposed a three-culture model consisting of MUTZ-3-derived LCs, KCs (HaCaT), and primary dermal fibroblasts to simulate metabolic environment, which may help predict the sensitization ability of pre/pro-haptens [56]. However, to date, no co-culture model of KCs and DCs can be used to evaluate the skin sensitization potential from a regulatory perspective [52]. The reconstructed human epidermis (RHE) model focuses on the environmental medium for skin cell growth. In this system, cutin cells are cultured under specific conditions, and the cutin layer is obtained by means of gas–liquid level (ALI) culture, which shows a good prospect in the screening of skin sensitizers. The RHE model has been applied to evaluate the irritancy potency of chemicals by determining the concentration that reduces the viability of epidermal equivalent by 50%, and that increases IL-1α secretion by 10-fold. This test cannot distinguish irritants and sensitizers but can achieve a 92% accuracy when combined with LLNA [57].

However, neither the co-culture nor the traditional RHE model can completely reflect the physiological complexity of human skin and accurately simulate the complex processes in the body. Therefore, the 3D culture model for skin sensitization has gradually attracted the interest of scientists. Chau et al. developed a 3D model, consisting of a layer of KCs, a layer of fibroblasts, and a central matrix of DCs (cultured in an agarose-fibronectin gel) [51]. The model can respond to low concentrations of stimuli and can be easily separated into a single layer for further study of cell–cell interactions. Van den Bogaard et al. proposed a model based on human skin equivalents, populated with CD4+T cells to study the interaction between KCs and T cells in a 3D microenvironment [58]. T cell migration into the dermis will initiate KCs activation within 2 days and cause the hallmarks of a psoriasiform inflammation after 4 days. Wei et al. selected some toxic compounds identified in 2D monolayer culture systems, and further tested their irritancy potency in 3D models of RHE and full thickness skin [59]. The secretion of IL-1α and IL-18 were used to evaluate the skin irritation of compounds. Overall, the 3D culture model is an area of growing interest in medical research and presents a broad development prospect in the study of the complex immune system.

### 4.6. Summary

At present, due to the complete prohibition of animal testing of cosmetic ingredients, recently, in vitro skin sensitizer screening methods have developed rapidly, many of which have been validated and applied. In the assays above, every KE in the AOP can be measured, and for every KE, suitable and stable cell lines have been applied with few but precise detection indicators. However, since skin sensitization is a complex process, any single alternative will not completely replace current animal testing, even those methods accepted as international testing guidelines. Therefore, it is necessary to integrate information from multiple alternative methods to overcome the limitations of separate tests and assess skin sensitization potential more accurately [60]. Even so, the development of skin sensitizer screening methods provides am efficient and reliable idea for other toxicity assays.

## 5. Cell Phenotype-Based Immunosuppression Assay

Unexpected immunostimulation is much less common than unexpected immunosuppression [61], and cases of immunostimulation are more likely to be drugs developed for immune deficiency [16]. In 2003, the European Centre for the Validation of Alternative Methods (ECVAM) proposed a hierarchical strategy that covered a myelotoxicity test, cell viability test and cell function test, in order to better identify the immunosuppression [20]. 

Nonetheless, studies on environmental immunotoxicity are mainly focused on direct immunotoxicity, including all changes that occur in immune system-related cells and their function or protein expression, without distinguishing between immunosuppression and immunostimulation. In these studies, functional analysis is often given priority in test, regardless of animal dependent or independent assays.

In the following section, we describe methods that have been, or have the potential to be, applied to immunotoxicity studies of environmental pollutants; however, several types of tests are excluded: (1) cytotoxicity of immune cells; (2) detection of reactions that are not unique to the immune system; (3) expression of genes or proteins that are only involved in mechanistic studies, but do not reflect cellular function. 

### 5.1. Animal-Dependent Assays

#### 5.1.1. In Vitro Antibody Response

In vitro antibody response, also known as Mishell–Dutton culture, was developed from a T cell-Dependent Antibody Response (TDAR) assay in rodents, the gold standard method for identifying immunosuppressive chemicals. In the standard TDAR experiment, animals are exposed to tested chemicals for 28 days and immunized with sheep red blood cells (SRBC) approximately 4 days before the end of exposure. Finally, levels of anti-SRBC Igs in serum were measured after exposure, to analyze whether immunosuppression occurred. The TDAR assay, based on the fact that antibody production by B cells requires involvement of both APCs and stimulated T cells, reflected abnormalities in specific immunity [3]. In the in vitro antibody response assay, mouse spleen cells were incubated for 5 days in the presence of tested chemicals and SRBC, then complement was added to dissolve SRBC in the antibody-secreting regions, resulting in hemolysis plaques, which represent antibody forming cells (AFCs) and reflect the activity of acquired immunity. This method avoids direct exposure to experimental animals and shortens the treatment time, showing great advantages [10]. However, as donor mice are still required, this method has never been considered as an alternative to animal experiments [62].

#### 5.1.2. Lymphocyte Proliferation Assay

Acquired immune response depends on the proliferation and differentiation of lymphocytes, including B cells and T cells, and inadequate proliferation will lead to immunosuppressive effect. Therefore, lymphocyte proliferation has been regarded as an indicator with great potential. Related experiments tend to use mixed lymphocyte, including spleen cells, thymocytes or lymph node cells from rodents, as well as lymphocytes isolated from human blood. In such an assay, lymphocytes are often stimulated by mitogen, which is selective and limited, such as phytohemagglutinin and concanavalin A to stimulate T cells and lipopolysaccharides (LPS) to stimulate B cells. When the targeted cell type is unclear, it is more efficient to use mitogens in combination or alone to act on both cell types [10]. Pokeweed mitogen is well received as a stimulus of both B and T cells [12]. For purified T lymphocytes, stimulus can also be a combination of anti-CD3 and anti-CD28 [20]. All mentioned above are nonantigenic stimuli, and another kind of stimulus, antigenic stimulus, includes staphylococcal enterotoxin B, tetanus toxoid and pneumococcal etc. [12]. Mixed lymphocyte reaction (MLR) is also an in vitro lymphocyte proliferation assay, with allogeneic cells as stimulus. In the absence of pretreatment, the reaction is bidirectional. If proliferation data of the specific cell is desired, stimulating cells can be irradiated with γ rays or treated with mitomycin C, before co-culture with the response cells. The proliferation ability of lymphocytes can be detected by tritiated thymidine ([3H]-TdR) incorporation, 5-Bromo-2’-deoxyuridine (BrdU) incorporation, MTT assay, and any other cell proliferation detection methods. Lymphocytes differentiation to subsets is often determined in conjunction with a proliferation assay, and the proportion of subsets can be obtained by flow cytometry to detect the expression of characteristic molecules on the cell surface [63]. For example, total T cells, Th and Tc can be analyzed by the expression levels of CD3, CD4 and CD8 molecules, respectively. CD16, CD19, CD2 and CD21 can be considered as the signature molecules of B cells [64].

### 5.2. Functional Assays of Certain Cells

#### 5.2.1. DC Maturation Assay

DCs undertake the important task of presenting antigens to T cells, greatly depending on DC maturation. Upon maturation, many changes occur, including the decreased phagocytosis function, increased ability of migration and the stimulation of T cell proliferation. Most of these changes are driven by the expression of characteristic molecules on the cell surface, including MHC II to presenting antigens, and co-stimulatory molecule (e.g., B7-1 (CD80), B7-2 (CD86), CD24, CD40) to activate T cells synergistically, chemokine receptors (e.g., CCR7, CXCR4) and down-regulation of E-cadherin to contributing to DC migration [65], and adhesion molecules (e.g., ICAM-1(CD54) and LFA-1(CD11a)) to promoting the binding of DCs to T cells [66]. Therefore, the expression levels of these molecules are often used as effective immunotoxicity indicators. 

Compared with detecting molecular expression, cell function assays provide more intuitive and convincing results, among which migration ability, phagocytosis and the ability to stimulate T cell proliferation are all important indicators. For example, Pinchuk et al. measured phagocytosis using fluorescein conjugated dextran and lucifer yellow to determine the highly selective mechanism and non-selective mechanism, respectively [67]. Vries et al. measured the cell adhesion rate by coating the bottom of the plate with fibronectin to quantify the migration capacity of DCs [68]. Transwell migration analysis is also utilized for the same target. Michalski et al. cultured DCs in the upper chamber and added chemokine CCL19 in the lower chamber to induce DCs to express the CCL19 receptor and migrate downward [65]. Then, DCs from the lower chamber were collected and measured to calculate migration rate. As for the detection of DCs’ ability to stimulate T cell proliferation, this is on the basis of co-culture with T cells. In such models, the endpoint is the reception of antigen signals from APCs by T cells, which can be quantified by measuring T cell proliferation after DC activation. The detection methods for T cell proliferation are just as used in the lymphocyte proliferation assay. In addition, Domogalla et al. showed that a tolerant DC (with low expression of co-stimulatory molecules and MHC) can cause high-proportioned Tregs and anergic T cells, which showed that T cell differentiation can also profoundly affect the immune response [69]. Therefore, the proportion of the T cell subpopulation should also be taken as an important indicator; this can be detected by identifying specific CD molecules. 

In such an assay, DCs can be obtained directly from cell lines (e.g., mouse JAWSII), or from monocytes derived from peripheral blood monocytes or bone marrow cultured with granulocyte-macrophage colony stimulating factor (GM-CSF), IL-4 and TNF-α [12]. Substances that can stimulate DC maturation include LPS (a natural TLR4 agonist), and Poly I:C (a synthetic double stranded RNA-like TLR3 agonist) can trigger DC maturation [67].

#### 5.2.2. Mφ Function Assay

Mφs and DCs can be derived from common precursor cells and have similar functions, including phagocytosis and antigen presentation, so their functions can be detected in the same way. However, for Mφs, phagocytosis is the more important function and the preferred indicator for immunotoxicity detection; it can be detected by fluorescently labeled pathogens or particles. For example, Li et al. incubated Mφs and fluorescence-labeled Latex beads for 5 h and calculated the phagocytic efficiency through intracellular fluorescence signal intensity [11]. Compared with fluorescence signal detection using flow cytometry, high-content screening (HCS) can effectively remove particles adsorbed on the cell surface, rather than being phagocytosed. Neaga et al. developed and validated a more stable approach using pH-sensitive pHrodo™ biological particles^®^, which can emit fluorescence only when taken up by low pH phagosomes [70]. In addition, Mφ phagocytosis depends on local actin cytoskeleton aggregation and significant changes in membrane morphology [71], but the morphological changes are difficult to quantify. Bitler et al. used atomic force microscopy (AFM) to image Mφs, and characterized morphological changes with fractal dimensions [72]. Recently, our team applied fluorescent markers containing phalloidin (a peptide that binds F-actin closely) to visualize the cytoskeleton, based on which we established an in vitro model of THP-1 derived Mφ [73]. Similar to DCs, the adhesion ability of Mφ is also considered to be a potential indicator for immune-mediated adverse reactions. Osma-Garcia et al. labeled Mφs with BCECF acetoxymethyl ester, and cultured them in plates previously coated with fibronectin [74]. Then, they measured the fluorescence intensity of all cells and remaining cells after removing nonadherent cells, and calculated cell adhesion rate.

As for the antigen presentation function, Mφs can only present antigens to activated T cells to maintain their function, which is different from the DC ability to activate naive T cells. For this reason, in the co-culture models of T cells and Mφs, T cell hybridomas are widely used to specifically recognize certain antigens. For example, Wang et al. explored the inhibitory effect of mycobacterium tuberculosis on Mφ antigen presentation [75]. Mφs derived from mouse bone marrow cells were first co-incubated with ovalbumin (OVA), then OVA-specific T-hybridoma cells were added for co-culture, and secret IL-2 as a final indicator. Compared with T cell lines, T cell hybridomas provide a chance for an animal-free antigen-presentation assay of Mφs, with extensive application prospects. By now, human antigen-specific T cell hybridomas have not been reported in antigen presentation studies. However, Bartley and Canaday have demonstrated that the adhesion molecules and integrins in human and mouse T cells are conservative in function, thus interaction between mouse T cells and human APCs can occur easily [76]. Therefore, the construction of a co-culture system containing mouse T cells and human APCs may be an important direction for an antigen-presentation assay of human APCs.

In relevant studies, Mφs are often obtained from the mouse abdominal cavity, lung and fish head kidney, etc. In addition, cell lines have been gradually applied, such as mouse Mφ cell line RAW264.7 and human mononuclear cell line THP-1, and the latter requires differentiation induced by phorbol ester (PMA) or GM-CSF.

#### 5.2.3. NKC Function Assay

Due to high sensitivity to toxic substances, NKCs are often used for chemical immunotoxicity in vivo assessment [12]. After exposure to the harmful substance, the proportion and activity of NKCs may change and, thus, increase the risk of cancer occurring. The NKC counts can be measured based on surface markers, but even more commonly used is the cytotoxic function assay. The Cytotoxic activity assay is based on the co-culture of NKCs and target cells (e.g., the classical primate K562 erythroleukemia cells or rodent YAC-1), during which target cells are killed and measurable substances are released. Such substances can be lactate dehydrogenase (LDH) [20], a kind of intracellular enzyme that normally cannot penetrate the cell membrane but will be released when membrane permeability changes, caused by the attack of NKCs. Another method utilizes target cells labeled with ^51^Cr, and detects the content of radioactive markers in culture supernatant for quantitative analysis [20]. In the process, it should be noted that the viability of target cells and amounts of radioactive markers released spontaneously in the absence of effector cells must be measured. Although the ^51^Cr release detection is a proven in vitro method to assess NKC activity, problems associated with radioactive material manipulation may hinder its use in many situations. Flow cytometry analysis is a safer and more environmentally friendly method, by staining target cells with a fluorescent dye that can penetrate the cell only when membrane integrity is damaged, so as to assess the cell mortality [77]. 

In the process of NKCs killing infected cells, perforin and cytotoxic granules play an indispensable role. When NKCs are activated, cytotoxic granules will quickly migrate to the immune synapse formed by NKCs with the target cell, and release their contents (e.g., granzyme). Perforin can form small pores in target cell membrane, leading granzyme into target cell cytoso, which induced apoptosis [78]. Therefore, cytotoxic function of NKCs can also be measured by detecting the levels of granulysin, perforin and granzyme [20].

#### 5.2.4. Cytokine Release Assay

As important molecules of intercellular communication, cytokines are often detected to analyze the function of various immune cells. The human whole-blood cytokine release assay (HWBCRA) is the only rigorous pre-validated immunotoxicity test designed to detect changes of IL-4 or IL-1β production, stimulated by SEB or LPS, but with the unresolved issue of individual difference [79]. Research on the cytokine release of human peripheral blood mononuclear cells (PBMCs) is also conducted. Compared with whole blood (WB), PBMCs’ lack of serum and red blood cells (RBC) are the real physiological conditions that may cause the change in the T cell cytokine response pattern, such as increased IL-1 and IL-2, decreased TNF-α and IFN-γ [80]. However, neither WB nor PBMC can be used to determine the directly affected cell. Therefore, cytokine release assay methods have begun to target cells of single type, including immune cells isolated from animals and cell lines. However, it is difficult to analyze the final effects caused by cytokines, due to high pleiotropy and redundancy of cytokines; this means that one cytokine may have multiple functions, and diverse cytokines may perform only one function. Therefore, it is appropriate to detect cytokines as widely as possible in in vitro systems [12]. In our study on the immunotoxicity of organophosphorus flame retardants (PFRs), a 48-plex panel kit was utilized to screen 48 targets expressed by THP-1 (human mononuclear cell line) derived Mφ simultaneously, and the results were more convincing [11]. Cytokine detection in most studies is conducted in a cell culture supernatant, unable to determine the proportion of cytokine-producing cells. Intracellular staining can analyze cytokine production in individual cells by adding a transport inhibitor to block the cytokine secretion and make up for the defects of traditional methods [81].

In addition to directly detecting the synthesis or secretion of cytokines, cytokine gene expression is also a manifestation of action of tested chemicals on cytokines, and has been applied to explore the influences of a tested substance on immune cells [82]. Genetics also offers another cytokine detection method, the reporter gene assay system, in which genes of luciferase or fluorescent protein are transferred into immune cell lines [83]; these are regulated in a similar way to cytokines, so that fluorescence intensity can indicate cytokine gene expression level. Kimura et al. utilized cell lines (THP-1 and Jurkat cell) transfected with luciferase to detect the expression of IL-2, IFN-γ, IL-1β, IL-8, and named this method the Multi-ImmunoTox Assay (MITA) [84]. Fluorescent protein has been used to develop the fluorescent cell chip (FCC), which is designed specifically for the fast screening of a large number of compounds. Due to the characteristics of high-throughput, both methods will present broad development prospects. 

### 5.3. Signaling Pathway Assay

Signaling pathway activation indicates multiple possible reactions with same or similar effects downstream, so signaling pathway assays are often conducted to study the mechanism of action for pollutants. In addition, assays targeting signaling pathways have also been developed. Immune signaling pathways start with PRRs, among which TLRs play a dominant role. There have been ten TLRs found expressing in the human body, present in cell membranes or endosomes to recognize different PAMPs, such as LPS (recognized by TLR4), lipopeptides (recognized by heterodimers of TLR2 with TLR1 or TLR6), flagellin (TLR5), single-stranded RNA (recognized by TLR7 and TLR8), double-stranded RNA (recognized by TLR3), and CPG-containing DNA (recognized by TLR9) [85,86]. TLRs, as important protein molecules to initiate innate immunity, can all activate the NF-κB pathway leading to transcription and the expression of a variety of pro-inflammatory cytokines. In addition, TLR7/8 and TLR9 can also activate IRF7 nuclear translocation, while TLR4 and TLR3 activate IRF3, both promoting the transcription of type I interferon. 

Since TLRs control the immune response of the host to pathogens, there have been many in vitro high-throughput drug screening methods targeting TLRs, which are often used for immunotherapy and vaccine development. Among these methods, some cell lines that stably express specific TLRs and are transfected with reporter genes are used, and they reflect the activation or inhibition of TLRs through the activity of enzymes transcribed by reporter genes. For example, Lan, T, et al. used HEK293 cells that stably express human TLR7 or TLR8, transiently transfected these with the secreted form of human embryonic alkaline phosphatase (SEAP) reporter plasmid, and then measured the SEAP activity of culture supernatants after exposure, to identify TLR-selective agonists [87]. Kokatla, H. P. et al. also utilized HEK293 cells, but these were stably co-transfected with hTLR, MD2, and secreted alkaline phosphatase (sAP). TLR agonists can induce sAP secretion under the control of NF-κB/AP-1 promoter, and the secretion amount reflects the induction of NF-κB [86]. 

In this signaling pathway, many other molecules are needed to transmit signals received by TLRs downstream. Specifically, the recognition of PAMPs stimulates the recruitment of a set of intracellular adaptors containing TIR-domains, including MyD88 and TIRAP (also known as MAL), Trif (also known as TICAM1) and TRAM (also known as TICAM2), through TIR-TIR interactions. The recruitment of these adapters can activate MAPKs (ERK, JNK, P38) and transcription factors (NF-κB, IRF3, etc.), controlling the expression of inflammatory cytokines, as well as type I IFN [85]. Currently, The Scripps Research Institute Molecular Screening Center has developed a high -throughput screening assay based on the combination of TLR and adaptors using a HeLa Cell Line (CL3-4). Such cells stably express two beta-lactamase (BLA) fragment fusion proteins: one fragment can be fused to MyD88 while the other can be fused to TLR4. The interaction of TLR4 and MyD88 will cause the two fragments of BLA to fuse together, making the cells show BLA activity, which can be reduced by interrupting the reconstruction of BLA [88]. Therefore, the BLA activity can reflect the activation of TLRs. On the whole, the proinflammatory cytokines are major promoters in the elimination of exogenous pathogens, in that they greatly affect the complex functions of various immune cells. Along this signaling pathway, NF-κB regulates the final transcription process. Therefore, Tox21 has developed an HTS method, which uses ME-180 cell lines stably expressing β-lactamase reporter genes under the regulation of NF-κB response elements, and the activity of β-lactamase will show the stimulation of chemicals to this pathway.

### 5.4. Summary

Currently, immunosuppression testing for environmental contaminants still relies on animals, including in vitro assays that need primary mixed or isolated cells. Compared with skin sensitization, there has not been a standard detection method and uniform detection index for immunosuppression; this has led to difficulty in comparing the immunotoxicity data of different substances and distinguishing among immunosuppressive, immunostimulatory or non-immunotoxic substances. For assays targeting signaling pathways, they can be used to largely predict the occurrence of a series of downstream events without detecting large quantities of downstream molecules. However, the difficulty of obtaining transfected cell lines actually hinders the application of these methods in the immunotoxicity detection of environmental pollutants. In addition, since different types of immunotoxic substances may act through different mechanisms, it is important to develop a corresponding detection framework for specific types of chemicals; however, this has not been elaborated in studies and may be limited by explicit mechanism study.

## 6. Inspirations for Paradigm Shift

### 6.1. Establishment of Immune AOP for HTS

AOP is a relatively new concept, and its goal is to link MIE directly with AO through a series of KEs to facilitate the exploration of certain toxicological mechanisms. An Adverse Outcome Pathway Knowledge Base (AOP-KB) has collected a total of 459 complete or incomplete AOPs, involving 5489 KEs [89] and including more than 20 immunity-related AOPs. Table 1 summarizes AOPs directly caused by immune system interference. MIEs initiating immune interference include receptor binding, protein binding and enzyme activity inhibition. These receptors are not limited to PRRs that are directly related to pathogen recognition, but ER-α and PPARγ stimulation can also cause disease-related immune function interference. In these limited AOPs, activation of the NF-κB signaling pathway, secretion of cytokines, recruitment of inflammatory cells, activation and differentiation of T cells, and the production of the cytokine are all important KEs in the mediation of the occurrence of some diseases. All of these have potential to be in the immunotoxic index. Table 2 summarizes immunity involved AOPs that are not caused by immune system injury. In these AOPs, it is endogenous molecules, known as damage-associated molecular patterns (DAMPs), released by damaged cells that initiate immune functions, mainly including increasing the recruitment/activation of inflammatory mediators and leukocytes. In this process, inflammatory mediators, which normally fight off invaders, will show adverse effects on the organism, indicating that the immune cells or molecules have dual characteristics, making immunotoxicity evaluation more difficult. These AOPs reflect the feature of the immune system that it is closely related to the systemic system, and immune related events are only partially involved in a portion of the AOPs of disease occurrence, further confirming that immunotoxicity is highly likely to be an early event of many diseases. For in vitro assay methods of immunotoxicity, attention needs to be focused on these frequently occurring immune-related KEs. In general, AO tends to be an organ, system or individual response. Since the immune system regulates systems throughout the body, any organ or tissue can be the subject of AO, but with similar immune cell effects. Therefore, it may be convenient to construct AOP networks by dividing different AOPs according to contaminant-induced cellular effects with the same immune tendency. For example, biased responses of Th1 and Th2 can lead to autoimmune diseases such as type 1 diabetes and rheumatoid arthritis; secretion of inflammatory cytokines and migration of inflammatory cells indicate the risk of triggering inflammation; and a decrease in T cell activity, total antibody production, cytotoxic function of NKCs, and phagocytic activity of Mφs may indicate the weakened immune function or immunosuppression. Such AOPs or AOP networks of immunosuppressive or immunostimulation are beneficial to link the effects of chemicals on immune cells, making the identification of immunotoxicity more comprehensive. On the other hand, since the MIE and every KE in AOP can become effective detection indicators, complete AOPs will further promote the construction of more stable, effective and convenient in vitro bioassay methods. Such methods have been established and applied for endocrine disrupters. For example, in the process of estrogen-like substances causing breast cancer in mammals [90], and damage on the reproductive system of fish [91], amphibians [92] and birds [93], ER activation and cell proliferation are considered as MIE and early KE, respectively [94]. However, at present, the research on pollutant immunotoxicity is so scarce that the mechanisms of action are still unclear, and that there are significant knowledge gaps for a complete AOP network. According to the experience of skin sensitization screening methods, it is still necessary to further search for MIEs and KEs associated with immunotoxicity, and to establish immune AOPs and standard in vitro assay methods based on MIEs and Kes to be applied to the HTS of chemicals.

### 6.2. Establishment of Functional Test Battery

In fact, the complex immune response cannot proceed normally without any kind of immune cells, and this results in an indeterminacy regarding whether damage to or a change in any single-type cells can indicate that immunosuppression is occurring [20]. To solve this problem, the test battery consisting of of the functional analysis of multi-type cells should be established. Valuable experience can be gained from genotoxicity test batteries, which cover different species, genetic endpoints and test methods, to reduce the incidence of false positives and improve the test sensitivity. However, the application of test batteries requires integrating several indicators though the combination of scientific evidence, interpretative methods, and expert judgment, which are known as the weight of evidence (WoE); this is a process in which several measurement endpoints are linked to a certain assessment endpoint to evaluate whether pollutants can pose a threat to an environment or human [95]. Due to the multi-function characteristic of immune cells, it is also a challenge to select simple but representative enough detection indicators for single-type cells.

In order to keep up with the development of high-throughput bioassay, establishing more in vitro assay methods will be a promising direction for chemical immunotoxicity research. There have been several functional assays described above, and the first task is to find a stable cell line that could replace animal cells. It is based on this idea that our team used THP-1-derived Mφs, detecting phagocytosis, adhesion, cell morphology and cytokine production, and applied it for the first time in the immunotoxicity detection of Yangtze River samples [96], providing a reference for the development of in vitro immunotoxicity detection methods. However, because of the complexity of the immune system, there is no single method that is completely accepted and standardized for pollutant assessment by OECD or EPA, resulting a in decrease in data comparability, and a difficulty in the toxicity prediction of an unknown material. In fact, T cells and B cells are the main force in acquired immunity, and the functional changes in them must be detected to determine whether immunotoxicity has occurred. It is apparent that effective animal-dependent immunotoxicity ex vivo assays, antibody response and lymphocyte proliferation target both T and B cell functions. However, such immune responses occur depending on interactions between different cells, and the function inhibition of one immune cell type may be compensated by enhanced function of another immune cell type. Therefore, a single type of immune cells cannot be used as a standard to judge whether immunotoxicity occurs, and functional analysis using more cell types (DCs, Mφs, NKCs, T cells and B cells) and co-culture models (DC/T, Mφ/T and B/T) may help address this obstacle.

### 6.3. Development of In Silico Methods

The increasingly growing number of publicly available toxicity databases and of valid and reliable data has provided an unprecedented favorable condition for the development of computational toxicology (in silico methods); this aims to predict toxicity, prioritize chemicals, guide toxicity tests, and minimize the use of animals [97]. Such methods that have been developed and applied include Structural Alerts and Rule-based Models, Chemical Category, Read-Across, Trend Analysis, and Quantitative Structure-Activity Relation-ship (QSAR) modeling [97]. Significant results have been obtained in the study of genotoxicity and endocrine disrupting effects. For example, electrophilic substances or non-electrophilic substances that produce electrophilic activity through metabolism are easy to react with nucleic acid. Additionally, some special groups or subunits of these substances are considered as structural alerts of genotoxicity, which facilitates the identification of genotoxic substances [98]. There are also many QSAR models of endocrine disruptors that have been established using methods including logistic regression, REPTree algorithms [99], generalized regression neural network [100], and genetic algorithm [101]. Generally, the data volume will no longer limit the development of in silico methods for genotoxicity and endocrine disrupting effects, because they can be supplemented by standardized in vitro methods. Immunotoxicity prediction, unfortunately, was conducted by few studies because of a small amount of data. 

Compared with immunosuppression, there is more research on in silico prediction of skin sensitizer; this also depends on more adequate data from in vivo/vitro experiments using the AOP-based sensitizer screening methods described above. Alves et al. proposed a comprehensive approach that integrates multiple QSAR models with in vitro, in chemico, in vivo, and human data into a Naive Bayes model for predicting human skin sensitization [25]. They also developed a web portal called Pred-Skin for the same purpose [102]. In addition, Russo et al. proposed a method for predicting ACD induced by skin sensitizers, using Universal Immune System Simulator (UISS), a state-of-the-art computational framework that can fully simulate the dynamic immune system [24]. It has also been shown to mimic the effects of the environmental immunotoxic substances PFAS on the immune system [103]. However, in contrast to skin sensitization, immunosuppression has no clear indicators or ways to determine its presence, which makes the selection of toxicity data very difficult. Schrey et al. used molecular fingerprints describing chemical structures as parameters of a machine learning approach, based on Naive-Bayes algorithms, and used cytotoxicity as immunotoxicity data to establish screening models [104]. However, it is questionable whether immunotoxicity can be estimated from the cytotoxicity of immune cells. In the above article, we focus on the functional analysis of immune cells, which should be based on the premise of exposure to non-cytotoxic concentrations of pollutants. However, functional analysis is not normative and there are few available data. In addition to cytotoxicity, Masi et al. pointed out that RACK1 may be used as a screening tool for the immunotoxicity of endocrine disrupters, based on the fact that hormones (e.g., glucocorticoids, estrogens and androgens) are involved in regulating the expression of RACK1 and the activation of immune cells simultaneously [105]. Karunarathne et al. used molecular docking methods to suggest that BPA may be a potential agonist for TLR4/MD2, which may trigger downstream signaling pathways of TLR4 and influence immune responses [106]. However, the reliability of this method needs to be further verified by in vitro experiments.

To sum up, an in silico method for immunotoxicity prediction is the key direction in the long run, but the first problem to be solved is the standardization of immunotoxicity in in vitro assay methods.

## 7. Conclusion and Prospects

Due to the importance and sensitivity of the immune system to pollutants, it is necessary to incorporate immunotoxicity into chemical risk assessments. However, the lack of relevant studies, and unclear molecular targets and modes of action, have greatly hindered the development of immunotoxicity research.

Driven by Tox21, it is urgent to find the nodes of immunotoxicity, identify effective indicators, and develop reliable in vitro assay methods to promote the paradigm shift in toxicity tests. At present, skin sensitization (an indirect immunotoxicity) has established standard assays relying on explicit AOP, providing guidance for the test of direct immunotoxicity. Currently, the assay methods for direct immunotoxicity involve complex and diverse indicators, but it is still unclear how to determine if immunotoxicity has actually occurred under the powerful compensation system. Thus, the AOP network and co-culture models must be constructed, and transcriptomics and proteomics can provide support for describing signaling pathways. In addition, in silico methods have shown a broad prospect, for which the in vitro assay methods must be developed first.

## Figures and Tables

**Figure 1 ijerph-20-00273-f001:**
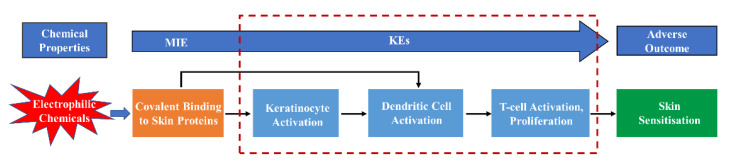
Adverse outcome pathways of skin sensitization.

**Table 1 ijerph-20-00273-t001:** Immune-related AOPs caused by immune system interference.

MIE		KE	AO
Macro-Molecular		Cell/Tissue	Organ/System
Receptor binding	Stimulation of TLR7/8	Maturation of DCs; Increase in IL-23 from matured DCs; Th17 cell migration and inflammation induction	Psoriatic skin disease
	Prolonged TLR9 activation	activation of MyD88; increase in inflammasome activity; Increase activation of NF-κB; Release of cytokine; Increase in pro-inflammatory mediators; Recruitment of inflammatory cells; Th17 cell migration and inflammation induction; A series of immune-related responses caused by DAMP release due to oxidative stress	ARDS and MODS;
	activation of TLR4	Activation of JNK; Activation of AP-1; Pin-1 activation; TGF-β expression; Activation of TGF-β pathway; Induction of EMT	Pulmonary fibrosis
	Inactivation of PPARγ	Activation of TGF-β signaling; Increase in Inflammation; Collagen Deposition; Induction of EMT	Lung fibrosis
	Binding to ER-α in immune cells	Induction of GATA3 expression; Increase in Th2 cells producing IL-4; Increase in autoantibody production	Exacerbation of SLE
Enzyme activity	Inhibition of JAK3	Blockade of STAT5 phosphorylation; Suppression of STAT5 binding to cytokine gene promoters; Suppression of IL-4 production	Impairment of TDAR
	Inhibition of calcineurin activity	Interference of nuclear localization of NFAT; Reduction of NFAT/AP-1 complex formation; Suppression of IL-2 and IL-4 production	Impairment of TDAR
Protein binding	Covalent Binding with Protein	Activation of keratinocytes; Activation, of dendritic Cells; Activation/Proliferation of T cells	Skin sensitization
	Covalent Binding with Protein	Activation of inflammatory cytokines, chemokines, cytoprotective gene pathways; Activation of dendritic Cells; Activation/Proliferation of T cells; Increased of proinflammatory mediator secretion	allergic respiratory hypersensitivity Response
Multifaceted action	Impairment of IL-1R1 signaling	Inhibition of NF-κB; Suppression of T cell activation	Increased of susceptibility to infection
	Interaction with the lung cell membrane	Increase in proinflammatory mediators; Recruitment of inflammatory cells; Loss of alveolar capillary membrane integrity; Activation of Th2 cells; proliferation and differentiation of Th2 cells; Increase in extracellular matrix deposition	Pulmonary fibrosis

Abbreviations: AO: adverse outcome; AP-1: Activator protein 1 activation; ARDS: Acute Respiratory Distress Syndrome; DC: dendritic Cell; EMT: Epithelial Mesenchymal Transition; ER: estrogen receptor; JNK: c-Jun N-terminal kinase; KE: key event; MIE: molecular initiating event; MODS: Multi Organ Failure. NFAT: nuclear factor of activated T cells; NF-κB: Nuclear factor κB; SAA: serum amyloid A protein; SLE: systemic lupus erythematosus; STAT: signal transducer and activator of transcription; TDAR: T cell-dependent antibody response; TGF: Transforming growth Factor; TLR: Toll-like receptor.

**Table 2 ijerph-20-00273-t002:** Immune-related AOP caused by non-immune system injury.

MIE	KE	AO
Macro-Molecular	Cell/Tissue	Organ/System
Sensing of the stressor by pulmonary cells	Increase in pulmonary pro-inflammatory cytokines; Increase in pulmonary SAA; Formation of HDL-SAA; Increase in systemic total cholesterol pool; Formation of foam cell	Plaque progression in arteries
Reactive Metabolite	Mitochondrial dysfunction; ER stress; Apoptotic cell death; Immune cell activation; IFN-γ signaling; Increase in inflammation	hepatitis
ACE2 enzymatic inhibition	Increased Angiotensin II; Binding of agonist, AT1R; Increase in ROS; Increase activation in NF-κB; Accumulation of collagen; Increase in proinflammatory mediators	Lung fibrosis
Alkylation of protein	Cell injury/death; Tissue resident cell activation; Increased pro-inflammatory mediators; Activation of stellate cells; Accumulation of collagen	Liver fibrosis
DNA damage; Increase in RONS	Tissue resident cell activation; Increased pro-inflammatory mediators; recruitment/activation of leukocytes; Increase in epithelia cell proliferation; Increase in mutations	Ductal Hyperplasia; Breast Cancer
Decrease in fibrinolysis; hyperactivation of bradykinin system	Increase in proinflammatory mediator secretion; Increase in inflammatory cells recruitment	Hyper inflammation
Endocytotic lysosomal uptake	Disruption of lysosome; Mitochondrial dysfunction; Cell injury/death; Increase in Pro-inflammatory mediators; recruitment/activation of Leukocyte; Activation of stellate cells; Accumulation of collagen	Liver fibrosis
Inhibition of IKK complex	Inhibition of NF-κB; Activation of caspase 8 pathway; Cell injury/death; Activation of Tissue resident cells (Kuppfer cells); Increase in proinflammatory mediators (TNF-α); Necrotic Tissue	Liver Injury

Abbreviations: ACE: angiotensin converting enzyme; AO: adverse outcome; AT1R: Angiotensin II receptor type 1 receptor; ER: endoplasmic reticulum; IFN: interferon; IKK: inhibitor of nuclear factor κB kinase; KE: key event; MIE: molecular initiating event; NF-κB: Nuclear factor κB; ROS: reactive oxygen species; RONS: reactive oxygen and nitrogen species; TNF: tumor necrosis factor.

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
