# Peer review of "Immunotoxicity In Vitro Assays for Environmental Pollutants under Paradigm Shift in Toxicity Tests"

_ijerph, 2022, doi:10.3390/ijerph20010273_

Round 1

Reviewer 1 Report

Introduction:

In row 73 authors report: "In recent years, our team has also been committed to improving current in vitro assay methods, establishing new methods and applying them to the detection of environmental samples, and has achieved remarkable results". I think they have to report the references demonstrating this results.

Methodology:

This paper is classified as a review, but the review method is not described. Is this a systematic review of a narrative review, or other? How the references have been selected?

Authors should dedicate a paragraph to describe the method of review used.

Conclusion:

In the conclusions, the final sentence (rows 791-795) introduces the possible application of QSAR model; since the authors not described it before, this conscept should be better diveloped and bibliographic references should be reported.

Author Response

Dear reviewer:

We greatly thank you for the time spent making useful advice, which is helpful to raise the quality of our manuscript. Our response to your questions is as follows, and other detailed modification information has been described in the cover letter.

Comment and Suggestion 1:

Introduction: In row 73 authors report: "In recent years, our team has also been committed to improving current in vitro assay methods, establishing new methods and applying them to the detection of environmental samples, and has achieved remarkable results". I think they have to report the references demonstrating this results.

Response 1:This suggestion is essential to make our descriptions more credible. We have inserted references that indicate our team's recent effects about the development and application of in vitro methods for detecting genotoxicity and endocrine disrupting in the introduction. 1 2 3 4

(1)   Ye, Y.; Weiwei, J.; Na, L.; Mei, M.; Donghong, W.; Zijian, W.; Kaifeng, R. Assessing of Genotoxicity of 16 Centralized Source-Waters in China by Means of the SOS/Umu Assay and the Micronucleus Test: Initial Identification of the Potential Genotoxicants by Use of a GC/MS Method and the QSAR Toolbox 3.0. Mutat Res Genet Toxicol Environ Mutagen 2014, 763, 36–43. https://doi.org/10.1016/j.mrgentox.2013.11.003.

(2)   Ye, Y.; Weiwei, J.; Na, L.; Mei, M.; Kaifeng, R.; Zijian, W. Application of the SOS/Umu Test and High-Content in Vitro Micronucleus Test to Determine Genotoxicity and Cytotoxicity of Nine Benzothiazoles. Journal of Applied Toxicology 2014, 34 (12), 1400–1408. https://doi.org/10.1002/jat.2972.

(3)   Jiang, W.; Yan, Y.; Ma, M.; Wang, D.; Luo, Q.; Wang, Z.; Satyanarayanan, S. K. Assessment of Source Water Contamination by Estrogenic Disrupting Compounds in China. Journal of Environmental Sciences 2012, 24 (2), 320–328. https://doi.org/10.1016/S1001-0742(11)60746-8.

(4)   Han, Y.; Li, N.; Oda, Y.; Ma, M.; Rao, K.; Wang, Z.; Jin, W.; Hong, G.; Li, Z.; Luo, Y. Evaluation of Genotoxic Effects of Surface Waters Using a Battery of Bioassays Indicating Different Mode of Action. Ecotoxicology and Environmental Safety 2016, 133, 448–456. https://doi.org/10.1016/j.ecoenv.2016.07.022.

Comment and Suggestion 2:

Methodology: This paper is classified as a review, but the review method is not described. Is this a systematic review of a narrative review, or other? How the references have been selected?

Response 2:This is not a systematic review but a narrative review. The focus of this paper is not the selection of literature, but the selection of assay methods for immunotoxicity detection in literature. These methods are derived from guidance documents of international organizations, scientific articles or databases. In addition to skin sensitization, for which a variety of standard methods have been developed, we focus on immune cell functional analysis for the detection of direct immunotoxicity. Therefore, several types of tests are excluded: (1) cytotoxicity of immune cells; (2) detection of reactions not unique to immunotoxicity, e.g. ROS production; (3) expression of genes or proteins that are only involved in mechanistic studies but do not reflect cellular function.

And I have already added some clarification in the last paragraph of the introduction.

In fact, due to the particularity of immune system, medical scientists have made far more contributions than environmental scientists in the study of immunotoxicity. We try to learn some methods or ideas from medicine or immunology and apply them to the environmental field. With the ultimate aim of developing high-throughput methods, this paper summarizes in vitro methods that have been or have the potential to be ap-plied to immunotoxicity studies of environmental pollutants. These methods derive from standard methods developed by international organizations (e.g. OECD), methods used in the scientific literature, and methods described in chemical databases (e.g. Pubchem), providing reference for the future application of immunotoxicity detection methods in the environmental field.”  

And in the begin of the section “5. cell phenotype-based immunosuppression assay”, I added some clarification as:

“In the following section, we described methods that have been or have the potential to be applied to immunotoxicity studies of environmental pollutants, but several types of tests are excluded: (1) cytotoxicity of immune cells; (2) detection of reactions that are not unique to immune system; (3) expression of genes or proteins that are only involved in mechanistic studies but do not reflect cellular function.”

Comment and Suggestion 3:

Conclusion: In the conclusions, the final sentence (rows 791-795) introduces the possible application of QSAR model; since the authors not described it before, this concept should be better developed and bibliographic references should be reported.

Response 3:This is really an important issue that we ignored, making our article incomplete. Therefore, I have added a subsection to illustrate the relevant methods in silico and indicate that there is a problem of insufficient data for the prediction of immunotoxicity at present, but other in silico methods for other toxicity have reference value for it.

Reviewer 2 Report

The paper shows an interesting topic, the bibliographic revision is complete, the paper is well structured, the description of the different methods looks clear, the only question is related to the indications of the different methods and its  priority for its application in the different  chemical compounds

Author Response

Dear reviewer:

We greatly thank you for the time spent making useful advice, which is helpful to raise the quality of our manuscript. Our response to your question is as follows, and other detailed modification information has been described in the cover letter.

Comment and Suggestion: The paper shows an interesting topic, the bibliographic revision is complete, the paper is well structured, the description of the different methods looks clear, the only question is related to the indications of the different methods and its priority for its application in the different chemical compounds.

Response:It is truly more reasonable to use specific methods or establish a priority level for detecting immunotoxicity or other toxicity of a specific type of compound, but this is based on the clear mechanism of toxicity of each type of chemical. The clarity of this mechanism should include the target cells or target molecules of the toxic effects of the chemical. However, for immunotoxicity, this is difficult for several reasons, which have been already stated in the original article .

  1. The immunotoxicity mechanisms of action of some substances are not well understood. For example, PCBs have been repeatedly reported to cause mass death in Marine mammals by affecting their immune systems. PCBs are often concerned as dioxin-like substances. Dioxins are thought to exert immunotoxicity mainly through AhR, while the immunotoxic effects of PCBs have been shown to be achieved through non-AhR dependent pathways, but the specific mechanism and target of action are unknown.
  2. Some immunotoxic substances have obvious effects on a variety of immune cells, and the effect on one type of cells will further affect another type of cells. But the questions that which type of cells are affected more, and that to what extent this effect contributes to related diseases, are unclear.
  3. In practice, no single in vitro method can completely predict immunotoxicity, and it may be appropriate to construct a test battery for a particular AOP or multiple cells, which has been described in the original article.

I have added a simple explanation in section 5.4 summary: “In addition, since different types of immunotoxic substances may act through different mechanisms, it is important to develop a corresponding detection framework for specific types of chemicals, but this has not been elaborated in studies, which may be limited by explicit mechanism study.